# Enhancing Transferable Adversarial Attacks on Vision Transformers through Gradient Normalization Scaling and High-Frequency Adaptation

**Zhiyu Zhu**[1], **Xinyi Wang**[2], **Zhibo Jin**[1], **Jiayu Zhang**[3] **& Huaming Chen**[1] *

University of Sydney[1], Universiti Malaya[2], SuZhouYierqi[3]

## Abstract

Vision Transformers (ViTs) have been widely used in various domains. Similar to Convolutional Neural Networks (CNNs), ViTs are prone to the impacts of adversarial samples, raising security concerns in real-world applications. As one of the most effective black-box attack methods, transferable attacks can generate adversarial samples on surrogate models to directly attack the target model without accessing the parameters. However, due to the distinct internal structures of ViTs and CNNs, adversarial samples constructed by traditional transferable attack methods may not be applicable to ViTs. Therefore, it is imperative to propose more effective transferability attack methods to unveil latent vulnerabilities in ViTs. Existing methods have found that applying gradient regularization to extreme gradients across different functional regions in the transformer structure can enhance sample transferability. However, in practice, substantial gradient disparities exist even within the same functional region across different layers. Furthermore, we find that mild gradients therein are the main culprits behind reduced transferability. In this paper, we introduce a novel Gradient Normalization Scaling method for fine-grained gradient editing to enhance the transferability of adversarial attacks on ViTs. More importantly, we highlight that ViTs, unlike traditional CNNs, exhibit distinct attention regions in the frequency domain. Leveraging this insight, we delve into exploring the frequency domain to further enhance the algorithm's transferability. Through extensive experimentation on various ViT variants and traditional CNN models, we substantiate that the new approach achieves state-of-the-art performance, with an average performance improvement of 33.54% and 42.05% on ViT and CNN models, respectively. Our code is available at: https://github.com/LMBTough/GNS-HFA.

## 1 Introduction

Vision Transformers (ViTs) has made significant strides in various domains, including image classification (Chen et al., 2021a), semantic segmentation (Strudel et al., 2021), and cross-modal tasks (Frank et al., 2021). However, current research suggests that ViTs are susceptible to adversarial samples, resulting in erroneous predictions (Shao et al., 2021). It is noteworthy that existing studies on adversarial attacks are primarily for Convolutional Neural Networks (CNNs) (Dong et al., 2018; Long et al., 2022; Zhang et al., 2022), and can't yield promising results when applied to ViTs (Bhojanapalli et al., 2021). Considering the positive outcomes of adversarial attacks for robustness assessment and model defense development, it is imperative to formulate effective adversarial attack methods for detecting vulnerabilities in ViT models in safety-critical applications.

Adversarial attacks are generally categorized as white-box and black-box attacks (Jin et al., 2024). In a white-box environment, the attacker has access to information about the target model's architecture or gradients. Conversely, for black-box attacks, such information is not available, and the attackers can solely rely on the model's inputs and outputs to construct adversarial samples. As one of the

---

*Corresponding author: huaming.chen@sydney.edu.au

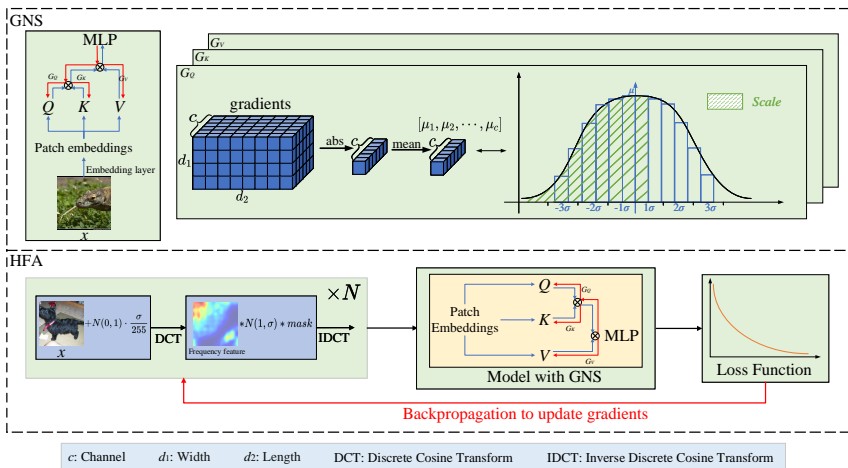

Figure 1: Illustration of our GNS-HFA method. Red links represent gradients backpropagation.

most effective black-box attack methods, transferable adversarial attacks can generate adversarial samples on the local proxy model and directly transfer them to attack the target model (Jin et al., 2023; Zhu et al., 2023; 2024). Considering the black-box nature of ViT-based applications, we herein emphasize the black-box attack, in particular transferable adversarial attacks, for ViT models.

However, there are challenges for applying transferable attacks to ViTs. On one hand, due to inherent structural differences between ViTs and traditional CNNs (Wei et al., 2022), the attack methods on CNNs (i.e., Spectrum Saliency map-based Attack (SSA) (Long et al., 2022) and Neuron Attribution-based Attack (NAA) (Zhang et al., 2022)) perform poorly on ViTs (see **Sec. 5.2**). On the other hand, current transferable attacks on ViTs aim to improve the sample transferability by fine-tuning gradients and applying gradient regularization within different functional regions of the transformer structure (Zhang et al., 2023; Wei et al., 2022). We observe that, even within the same functional region, there exists a variance in the impact of gradients from different channels on sample transferability. In contrast, existing methods focus on an entire functional region, such as an entire attention layer or MLP layer, which results in limited transferability. Specifically, the Token Gradient Regularization method (TGR) (Zhang et al., 2023) regularizes extreme gradients (see **Sec. 4.1**) within the same layer to enhance transferability. Pay No Attention method (PNA) (Wei et al., 2022) optimizes adversarial samples by discarding a certain number of gradients within the same layer. Yet, as different layers capture diverse features and information, regularizing or removing gradients from specific layers may cause significant information loss. Additionally, simply discarding the gradient can make the algorithm more susceptible to data perturbations or noise, impacting attack robustness.

To solve the challenges, in this paper, we present a novel Gradient Normalization Scaling (GNS) method for fine-grained gradient editing. Through gradient normalization scaling, we scale mild gradients (see **Sec. 4.1**) in the backpropagation process to prevent overfitting during the training of adversarial samples. Moreover, inspired by SSA (Long et al., 2022), we observe that ViTs exhibit significant attention to high-frequency features through attribution visualization (Pan et al., 2021). Thus, we propose the High-Frequency Adaptation (HFA) method to explore the sensitivity of ViTs to adversarial attacks in different frequency regions. Specifically, we construct a mask to differentiate between high-frequency and low-frequency information, allowing for targeted explorations of ViTs' performance in high-frequency regions. Fig. 1 presents the structure of our approach. We evaluate the proposed GNS-HFA method on various ViT variants and CNN models. Notably, compared with the state-of-the-art methods, our approach achieves an average improvement of 33.54% and 42.05% both on ViT and CNN models. Our contributions in this paper are summarized as follows:

**1.** We observe that in the gradient backpropagation, mild gradients have a significant impact on sample transferability. By accurately normalizing and scaling the identified mild gradients, we alleviate overfitting caused by gradient variations during the backpropagation process.

**2.** Noting the phenomenon that ViTs pay more attention to high-frequency regions, we propose the high-frequency adaptation method to stabilize the gradient update direction.

**3.** We present a novel Gradient Normalization Scaling and High-Frequency Adaptation(GNS-HFA) method to enhance the transferability of adversarial samples on ViTs.

**4.** We conduct extensive experiments to showcase the superiority of our approach compared to state-of-the-art transferable attacks on various ViTs and CNN models, achieving an average improvement of 33.54% and 42.05%. We also release the replication package of our GNS-HFA.

## 2 RELATED WORK

### 2.1 VISION TRANSFORMER

Vision Transformer (ViT) (Dosovitskiy et al., 2020) is a type of deep neural network architecture based on self-attention, initially used in natural language processing but now widely applied in computer vision. Compared to traditional CNNs and RNNs, it offers higher representational capacity and fewer visual-specific biases, showing similar or superior performance in visual benchmarks (Han et al., 2022). This has led to the proposal of numerous ViT models.

ViT-B/16 (Dosovitskiy et al., 2020) is tailored for image classification, processing images in fixed-size blocks as sequential inputs. PiT-B (Heo et al., 2021) employs pooling to enhance its capability, reducing spatial dimensions while increasing channel dimensions for improved information processing. CaiT-S/24 (Touvron et al., 2021b) deepens the Transformer and introduces classification labels for enhanced image classification performance. Visformer-S (Chen et al., 2021b) transforms the Transformer into a convolution-based model, combining the advantages of both. DeiT-B (Touvron et al., 2021a) is another Transformer-based model for image classification, effectively processing image data for improved performance. TNT-S (Han et al., 2021) utilizes 'cross-shaped window self-attention' for efficient analysis of different image parts, enabling quick processing of large-scale images while maintaining high performance. LeViT-256 (Graham et al., 2021) is a lightweight ViT with high classification accuracy and lower computational costs, ideal for resource-constrained environments. ConViT-B (d'Ascoli et al., 2021) combines convolution and Transformer strengths for image classification, employing a hybrid model structure to achieve improved performance.

### 2.2 TRANSFERABLE BLACK-BOX ADVERSARIAL ATTACKS

Currently, transferable black-box adversarial attacks can be categorized into three main groups: feature-level attack methods, input transformation methods, and advanced gradient methods. Feature-level attack methods such as Neuron Attribution-based Attack (NAA) (Zhang et al., 2022), Feature Importance-aware Attack (FIA) (Wang et al., 2021) and Feature Disruptive Attack (FDA) (Ganeshan et al., 2019) aim to find accurate estimates of the importance of intermediate-layer neurons to perform transferable attacks. Input transformation methods involve transforming input image samples to enhance their transferability. Diverse Input Method (DIM) (Xie et al., 2019) introduces multiple randomly generated gradient directions and perturbs the samples along different directions to increase the diversity of adversarial samples. Translation Invariant Method (TIM) (Dong et al., 2019) introduces translation invariance, generating adversarial samples with similar perturbations at different positions and scales. Since numerous studies have underscored the substantial correlation between DNNs and the frequency domain (Wang et al., 2020; Guo et al., 2018; Yin et al., 2019), Spectrum Saliency Attack (SSA) (Long et al., 2022) explores the impact of gradients on DNNs in the frequency domain to craft more transferable samples, addressing the limitations associated with spatial domain-only model augmentation.

Unlike input transformation methods and feature-level attack methods, gradient based methods enhance sample transferability by employing gradient update techniques. Momentum Iterative Method (MIM) (Dong et al., 2018) introduces momentum to stabilize the gradient update direction during iterative perturbation updates. Scale Invariance and Nesterov Iterative Method (SINI-FGSM) (Lin et al., 2019) utilize the Nesterov accelerated gradient technique to update perturbations while incorporating a scale-invariant attack approach to effectively target images with diverse scales. TGR (Zhang et al., 2023) and PNA (Wei et al., 2022) scale or remove gradients from the same layer to assess the transferability of adversarial samples on ViT models. However, both TGR and PNA

have shortcomings in accurately locating and regularizing the gradients which may lead to overfitting during the backpropagation process.

# 3 PRELIMINARIES

## 3.1 PROBLEM DEFINITION OF TRANSFERABLE ATTACKS

Given the source model $f_\theta$, the initial sample $x$, and the target class $y$, in adversarial attacks, the goal is to discover a perturbation $\eta^*$ that maximizes the objective function $L\left(f_\theta(x + \eta), y\right)$, while satisfying the constraint $\|\eta\|_p \leq \epsilon$. It is essential to regulate the magnitude of the perturbation to prevent noticeable visual distortions resulting from adversarial manipulations. To achieve this, we apply constraints on the perturbation $\eta$ using $L_p$ norms. Here, $\epsilon$ is a predefined upper bound representing the maximum perturbation size. Consequently, our attack objective can be expressed as an optimization problem in Eq. 1:

$$\eta^* = \arg\max_\eta L\left(f_\theta(x + \eta), y\right) \text{ s.t. } \|\eta\|_p \leq \epsilon \tag{1}$$

$$\max L\left(f_{\theta_i}(x + \eta^*), y\right), f_{\theta_i} \in \{f_{\theta_1}, f_{\theta_2}, ..., f_{\theta_n}\} \tag{2}$$

In the context of transferable attacks, as illustrated in Eq. 2, once the perturbation rate $\eta^*$ is determined through Eq. 1, our objective is to introduce perturbations generated on $f_\theta$ to magnify the loss functions of other black-box models. It's important to emphasize that the $f_{\theta_i}$ in Eq. 2 is not equivalent to the $f_\theta$ in Eq. 1. While these variables share similar goals, they possess distinct underlying structures. More specifically, we generate adversarial samples on the surrogate model denoted as $f_\theta$, aiming to execute successful adversarial attacks on the black-box models represented by $f_{\theta_i}$ to enhance the transferability of adversarial attacks across all black-box models.

## 3.2 MULTI-HEAD SELF-ATTENTION IN VITS

In ViT models, the Multi-head Self-Attention mechanism (MSA) allows the model to simultaneously attend to different parts of the sequence in different representation spaces, thereby improving the model's performance (Dosovitskiy et al., 2020). The key idea is to map the input sequence into multiple distinct subspaces and perform a self-attention operation in each subspaces. This enables the model to extract information from various perspectives or representations and then integrate this information to generate the final output. MSA typically involves the following key steps:

**Initial projection:** The input sequence is first mapped into multiple subspaces. Each subspace has its own weight matrices, typically denoted as $Q$ (queries), $K$ (keys), and $V$ (values).

**Attention calculation:** For each subspace, self-attention calculations are performed. This involves computing similarity scores between queries and keys and using these scores as weights to average the value vectors, resulting in attention outputs for each subspace.

**Multi-head combination:** The attention outputs from multiple subspaces are combined, often by concatenating or adding them along a specific dimension.

**Final projection:** Finally, the merged attention outputs are projected again through a projection layer to produce the ultimate multi-head self-attention output.

# 4 METHOD

## 4.1 GRADIENT NORMALIZATION SCALING (GNS)

**Feasibility of gradient fine-grained editing** For CNN models, there exists a similarity in features between the surrogate model and the target model. Leveraging this similarity, we can generate transferable samples from the surrogate model. However, transferable attacks on ViT models often exhibit the phenomenon of overfitting, wherein the attack success rate is high on surrogate models but low on the target model. TGR (Zhang et al., 2023) argues that such a phenomenon is caused by extreme gradients during backward propagation, and removing these extreme gradients can significantly enhance sample transferability. Specifically, extreme gradients are defined as tokens whose

backpropagated gradient magnitudes rank in the top-$k$ or bottom-$k$ among all tokens, with $k$ being a hyperparameter. Obviously, the key for the surrogate model to produce transferable samples lies in those relatively large gradients. Therefore, if these extreme gradients are scaled or clipped, the attack capability of samples will change dramatically. To strike a balance between mitigating overfitting and retaining attack capability, TGR directly sets unwanted extreme gradients to zero, which diminishes the involvement of gradients that possess attack capability. In this paper, we find that the extreme gradients in TGR are not the main culprits behind reduced transferability. Instead, as shown in Fig. 3, the occurrence of overfitting primarily resides in relatively small gradients (Further discussed in **Appendix**). These gradients, which we term as mild gradients, have minimal impact on attack capability. Therefore, they can be extensively scaled.

Also, an intuitive explanation of why mild gradients are more likely to cause overfitting is that, due to the differing internal structures of ViTs and CNNs, ViTs employ self-attention mechanisms to capture both global and local information in images. This mechanism renders the model highly sensitive to subtle perturbation in inputs during the generation of adversarial samples. If gradients are too small, the update step is relatively small, making the model excessively sensitive to small changes in the training data, thus making overfitting more likely and affecting the quality of locally trained adversarial samples. On the other hand, for deeper ViT models, mild gradients may also contribute to the vanishing gradient problem, preventing effective updates to the lower-level attention mechanisms and thereby affecting the training process of adversarial samples.

**Precise scaling of mild gradients** In ViTs, the primary mechanism for handling features is the Multi-head Self-Attention structure. We perform fine-grained editing on the gradient information produced by the key parameters in the MSA structure, namely $W^Q$, $W^K$, and $W^V$. It is worth noting that gradient information has the same dimensionality as the parameters. We can decompose the gradients into $G = \{g_1, g_2, ..., g_C\}$, where $C$ represents the number of channels (corresponding to the number of channels in the convolution kernel or the number of multi-head attention), and it can be considered that each channel is independent (Zhang et al., 2023). Since the sign information can only represent the direction in the gradient, and does not determine the size of the gradient itself, we use the absolute value to ignore the directional information. To accurately identify the gradients to be scaled, we compute the absolute mean of the gradients for each channel, as shown in Eq. 3:

$$
\begin{aligned}
g_{abs\_mean} &= \left[\ \mu_1, \mu_2, \cdots, \mu_c\ \right] \\
&= \left[\ mean(abs(g_1)), mean(abs(g_2)), \cdots, mean\left(abs\left(g_C\right)\right)\ \right]
\end{aligned}
\tag{3}
$$

Here, $abs(\cdot)$ represents the absolute value. Then, we identify the channel that requires scaling:

$$
\text{index } = \{l : \mu_l < \mu + u * \sigma\}
\tag{4}
$$

We define gradients with magnitudes less than $\mu + u * \sigma$ as mild gradients. Where $\mu$ represents the average value across $C$ channels. $\sigma$ represents the standard deviation across $C$ channels, and the hyperparameter $u$ denotes the allowed deviation level. By adjusting the hyperparameter $u$, we can correspondingly adjust the range of mild gradients. It's worth noting that our ablation experiments prove that the choice of $u$ is very general, and in most cases, excellent results can be achieved without the need for hyperparameter adjustment.

To distinguish which gradients in a chosen channel require scaling, we compute the deviation $abs(\frac{g_l - \mu}{\sigma})$ from the mean gradient for mild gradients. Scaling, as described in Eq. 5, is necessary due to the susceptibility of mild gradients to overfitting.

$$
g_l = g_l * tanh\left(abs\left(\frac{g_l - \mu)}{\sigma}\right)\right)
\tag{5}
$$

here we employ the $tanh$ activation function to map deviation values. If gradient information exhibits significant deviation, then the relatively large gradients play a crucial role in attack capability and should not be removed. Therefore, using $tanh$ activation function allows for the adaptive scaling of gradients. Alg. 1 shows the specific process of our GNS method.

## 4.2 HIGH-FREQUENCY ADAPTATION (HFA)

**Exploring high-frequency features through masking** Inspired by SSA (Long et al., 2022), in order to further enhance the transferability of adversarial samples on ViTs, we employ frequency

---

**Algorithm 1** GNS method

---

1: **Input:** gradient set $G$, deviation level $u$
2: $G = [\ g_1, g_2, \cdots, g_C\ ]$
3: $g_{abs\_mean} = [\ \mu_1, \mu_2, \cdots, \mu_c\ ] = [\ mean(abs(g_1)), mean(abs(g_2)), \cdots, mean\,(abs\,(g_C))\ ]$

4: $\mu = mean(g_{abs\_mean})$
5: $\sigma = \text{std}(g_{abs\_mean})$
6: index $= \{l : \mu_l < \mu + u * \sigma\}$
7: **for** $l$ in index **do**
8: $\quad g_l = g_l * tanh\left(abs\left(\frac{g_l - \mu}{\sigma}\right)\right)$
9: **end for**
10: **return** $G$

---

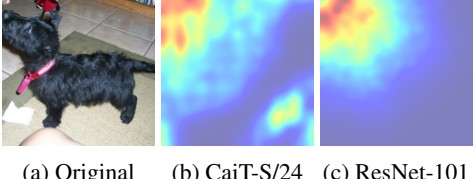

(a) Original    (b) CaiT-S/24    (c) ResNet-101

Figure 2: Attribution visualization (Pan et al., 2021) for different models in frequency domain

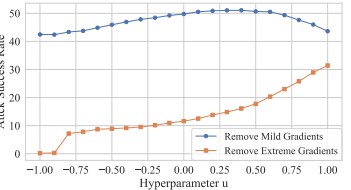

Figure 3: Performances of removing mild gradients and extreme gradients on ViT-B/16

exploration to optimize the gradient update direction in ViTs. As shown in Fig. 2, we observe that, compared to traditional CNN models, ViTs tend to focus more on high-frequency features. Therefore, we need to conduct targeted explorations of ViTs in high-frequency regions.

$$mask_{ij}^k = \frac{\left(\frac{W+i}{2}\right) \cdot \left(\frac{H+j}{2}\right)}{W \times H} \quad k = 0, 1, 2 \quad mask \in R^{W \times H \times 3} \tag{6}$$

$$x_f = IDCT\left(DCT(x + N(0,1) \cdot \frac{\epsilon}{255}) * N(1,\sigma) * mask\right) \tag{7}$$

In the frequency representation of the image, features near the top-left corner represent low-frequency features, while features near the bottom-right corner represent high-frequency features. In order to distinguish image features in different frequency regions, as shown in Eq. 6, we construct a mask and adjust the frequency exploration level based on different coordinate points. Here $i = 0, 1, ..., W; j = 0, 1, ..., H$. The image is 3-channel, where $W$ and $H$ represent the width and height of the image. As shown in Eq. 7, after multiplying with the corresponding noise, we obtain the expression of the features in the frequency domain. DCT represents the discrete cosine transformation method proposed by Ahmed et al. (Ahmed et al., 1974), which is a general method for frequency transformation. $\epsilon$ represents the perturbation magnitude. It is worth noting that if the original model has limited high-frequency information, the frequency exploration through multiplication may become ineffective. In such cases, it becomes necessary to introduce basic high-frequency information by adding and normalizing additional noise $N(0,1) \cdot \frac{\epsilon}{255}$.

**Transferable attack direction adaptation** In Eq. 8, multiple frequency features generate a more transferable gradient update direction when computed using the GNS-trained model for $\frac{\partial L(x_f, y)}{\partial x_f}$.

$$P_i = \frac{\partial L\,(x_f, y)}{\partial x_f}$$
$$x^{t+1} = x^t + \eta \cdot sign\left(\frac{1}{N}\sum_{i=1}^{N} P_i\right) \tag{8}$$

It should be noted that we randomly select $N$ backpropagation frequency gradients and employ their average value to update the adversarial sample in the current iteration. $P_i$ represents the gradients of the GNS-trained model relative to the frequency features $x_f$. $y$ represents the original label. $\eta$ represents the learning rate. Here $sign(\cdot)$ determines the gradient update direction. HFA utilizes

---

**Algorithm 2** HFA method

---

1: **Input:** input feature $x$, number of frequency samples explored $N$, perturbation magnitude $\epsilon$, learning rate $\eta$, number of iterations $T$
2: $mask_{ij}^k = \frac{\left(\frac{W+i}{2}\right) \cdot \left(\frac{H+j}{2}\right)}{W \times H} \quad k = 0, 1, 2 \quad mask \in R^{W \times H \times 3}$
3: $x^0 = x$
4: **for** $t$ in $range(T)$ **do**
5:     **for** $i$ in $range(N)$ **do**
6:        $x_{f_i}^{t+1} = IDCT\left(DCT(x^t + N(0,1) \cdot \frac{\epsilon}{255}) * N(1, \sigma) * mask\right)$
7:        $P_i^{t+1} = \frac{\partial L\left(x_{f_i}^{t+1}, y\right)}{\partial x_{f_i}^{t+1}}$ , using the GNS-trained model to calculate $\frac{\partial L\left(x_{f_i}^{t+1}, y\right)}{\partial x_{f_i}^{t+1}}$
8:     **end for**
9:     $x^{t+1} = x^t + \eta \cdot sign\left(\frac{1}{N} \sum_{i=1}^N P_i^{t+1}\right)$
10: **end for**
11: **return** $x^{t+1}$

---

the gleaned high-frequency information to adjust the update process of adversarial samples. This ensures that the generated adversarial examples better conform to the ViTs structure, thus can more stably cross the decision boundaries of target models. Alg. 2 shows the specific process of HFA.

## 5 EXPERIMENTS

In this section, we conduct our experimental setup following the literature (Zhang et al., 2023) to facilitate a fair comparison of various attack methods. Extensive analyses are performed on both ViT and CNN models. The details of our experimental setup are as follows.

### 5.1 EXPERIMENTAL SETUP

**Baselines:** Nine methods are selected as competitive baselines in this experiment, among which two transferable adversarial attack methods, TGR (Zhang et al., 2023) and PNA (Wei et al., 2022), are developed specifically for ViT model advancements. The remaining methods, SSA (Long et al., 2022), BIM (Kurakin et al., 2018), PGD (Madry et al., 2017), DI-FGSM (Xie et al., 2019), TI-FGSM (Dong et al., 2019), MI-FGSM (Dong et al., 2018), and SINI-FGSM (Lin et al., 2019), are optimized for CNN models. TGR is chosen as the primary competitive baseline against our method.

**Dataset:** In this experiment, we employ a dataset that is entirely consistent with our primary competitive baseline, TGR (Zhang et al., 2023), and currently the best-performing transferable attack method on CNNs, SSA (Long et al., 2022). This dataset comprised randomly selected 1000 images from the ILSVRC 2012 validation set (Russakovsky et al., 2015).

**Models:** In this experiment, a substantial number of models are selected to align completely with TGR, encompassing eight ViT models and seven CNN models. The ViT models included LeViT-256 (Graham et al., 2021), PiT-B (Heo et al., 2021), DeiT-B (Touvron et al., 2021a), ViT-B/16 (Dosovitskiy et al., 2020), TNT-S (Han et al., 2021), ConViT-B (d'Ascoli et al., 2021), Visformer-S (Chen et al., 2021b), and CaiT-S/24 (Touvron et al., 2021b). Among them, ViT-B/16, Visformer-S, PiT-B, and CaiT-S/24 are used as surrogate models to train transferable samples. The CNN models are Inception-v3 (Inc-v3) (Szegedy et al., 2016), Inception-v4 (Inc-v4) (Szegedy et al., 2017), Inception-ResNet-v2 (IncRes-v2) (Szegedy et al., 2017), and ResNet-101 He et al. (2016). Additionally, three models with enhanced adversarial robustness through ensemble training are selected: the ensemble of three adversarial trained Inception-v3 (Inc-v3-adv-3) (Tramèr et al., 2017), the ensemble of four adversarial trained Inception-v3 (Inc-v3-adv-4) (Tramèr et al., 2017), and Inception-Resnet-v2 (IncRes-v2-adv) (Kurakin et al., 2016).

**Evaluation Metrics:** We choose to employ the Attack Success Rate (ASR) as the evaluation metric for different attack methods. ASR calculates the proportion of samples in the dataset for which the adversarial attack method successfully misleads the model into classifying them incorrectly. Therefore, a higher ASR value indicates a stronger attack capability of the method.

Table 1: ASR on ViT and CNN Models. The table includes various adversarial attacks on different surrogate models including ViT and CNN architectures. The best results are in bold (%).

| Surrogate Models | Method | ViT | | | | | | | | CNN | | | | | | |
|---|---|---|---|---|---|---|---|---|---|---|---|---|---|---|---|---|
| | | LeViT-256 | PiT-B | DeiT-B | ViT-B/16 | TNT-S | ConViT-B | Visformer-S | CaiT-S/24 | Inc-v3 | Inc-v4 | IncRes-v2 | ResNet-101 | Inc-v3-adv-3 | Inc-v3-adv-4 | IncRes-v2-adv |
| ViT-B/16 | TGR | 65.60% | 55.70% | 88.00% | 99.60% | 80.40% | 88.40% | 62.50% | 86.60% | 55.40% | 50.60% | 45.20% | 51.30% | 38.80% | 38.40% | 33.20% |
| | SSA | 59.90% | 59.20% | 82.40% | 99.80% | 76.70% | 83.70% | 62.30% | 83.40% | 62.00% | 59.60% | 56.60% | 58.20% | 53.40% | 53.90% | **50.80%** |
| | PNA | 42.10% | 41.80% | 72.30% | 94.00% | 59.00% | 71.30% | 43.10% | 71.70% | 37.40% | 35.30% | 28.60% | 33.80% | 24.20% | 23.40% | 17.80% |
| | BIM | 14.30% | 14.80% | 36.00% | 100.00% | 26.50% | 39.10% | 16.30% | 38.20% | 13.10% | 10.60% | 10.50% | 11.60% | 7.90% | 5.90% | 5.40% |
| | PGD | 12.80% | 12.50% | 31.60% | 100.00% | 22.60% | 34.10% | 14.20% | 33.20% | 12.90% | 10.10% | 9.40% | 12.80% | 6.40% | 4.30% | 3.40% |
| | DI-FGSM | 34.70% | 37.50% | 55.00% | 98.30% | 49.00% | 59.60% | 37.50% | 58.70% | 29.90% | 30.00% | 25.50% | 26.70% | 22.00% | 21.40% | 17.40% |
| | TI-FGSM | 16.70% | 19.90% | 29.60% | 97.40% | 31.30% | 34.40% | 23.30% | 30.50% | 18.80% | 18.60% | 12.80% | 16.90% | 15.90% | 17.40% | 14.20% |
| | MI-FGSM | 34.40% | 33.90% | 62.70% | 99.90% | 51.20% | 64.20% | 36.60% | 64.70% | 33.20% | 30.50% | 25.90% | 32.30% | 23.60% | 21.10% | 18.90% |
| | SINI-FGSM | 45.70% | 39.00% | 75.30% | 100.00% | 65.80% | 76.50% | 45.10% | 77.60% | 46.00% | 44.40% | 36.50% | 43.10% | 36.60% | 36.50% | 31.10% |
| | GNS-HFA (Ours) | **76.80%** | **70.60%** | **93.50%** | 99.80% | **87.60%** | **92.50%** | **72.70%** | **92.40%** | **67.30%** | **64.10%** | **59.00%** | **63.10%** | **54.50%** | **55.80%** | 48.20% |
| Visformer-S | TGR | 79.10% | 71.50% | 65.70% | 43.50% | 79.50% | 58.00% | 100.00% | 67.80% | 76.30% | 75.90% | 65.70% | 72.40% | 45.00% | 38.90% | 28.80% |
| | SSA | 75.60% | 73.70% | 74.90% | 64.10% | 77.70% | 73.80% | 97.20% | 75.40% | 77.60% | 76.90% | 74.30% | 74.90% | 70.00% | 69.30% | 65.90% |
| | PNA | 65.80% | 61.90% | 46.90% | 28.80% | 69.10% | 44.40% | 100.00% | 52.40% | 53.30% | 53.20% | 40.70% | 45.70% | 23.70% | 19.90% | 15.40% |
| | BIM | 24.50% | 27.20% | 14.10% | 9.40% | 29.70% | 16.70% | 99.90% | 16.10% | 19.80% | 19.40% | 13.30% | 16.40% | 8.30% | 6.60% | 4.60% |
| | PGD | 26.80% | 24.20% | 14.20% | 10.90% | 27.20% | 14.60% | 99.90% | 15.10% | 20.70% | 20.90% | 14.10% | 17.20% | 7.30% | 5.80% | 4.40% |
| | DI-FGSM | 54.50% | 56.00% | 39.20% | 22.30% | 57.10% | 39.60% | 98.80% | 45.10% | 47.20% | 47.90% | 35.70% | 39.40% | 22.40% | 17.30% | 12.30% |
| | TI-FGSM | 26.70% | 34.70% | 27.30% | 19.90% | 38.90% | 29.60% | 95.00% | 29.30% | 28.60% | 28.00% | 19.50% | 21.80% | 19.30% | 21.80% | 16.70% |
| | MI-FGSM | 48.90% | 50.70% | 37.30% | 29.30% | 52.70% | 38.90% | 99.90% | 40.50% | 44.00% | 43.20% | 36.70% | 39.30% | 24.40% | 21.50% | 16.20% |
| | SINI-FGSM | 68.00% | 66.90% | 58.30% | 43.10% | 72.00% | 58.20% | 100.00% | 60.10% | 63.50% | 63.20% | 55.00% | 58.40% | 40.30% | 36.70% | 30.10% |
| | GNS-HFA (Ours) | **94.90%** | **92.20%** | **91.10%** | **80.90%** | **94.60%** | **89.60%** | 100.00% | **91.70%** | **95.30%** | **95.40%** | **92.70%** | **93.20%** | **89.60%** | **85.40%** | **80.20%** |
| PiT-B | TGR | 87.80% | 100.00% | 83.20% | 65.40% | 90.50% | 82.40% | 88.50% | 82.90% | 80.00% | 73.50% | 69.30% | 71.90% | 51.10% | 51.50% | 40.50% |
| | SSA | 64.20% | 94.90% | 66.90% | 59.20% | 71.00% | 66.50% | 67.10% | 66.00% | 63.80% | 64.50% | 59.30% | 58.90% | 55.20% | 55.10% | 51.80% |
| | PNA | 62.20% | 99.80% | 54.60% | 38.90% | 67.00% | 56.10% | 70.50% | 55.70% | 51.40% | 47.80% | 41.80% | 42.10% | 25.70% | 22.70% | 16.60% |
| | BIM | 17.60% | 100.00% | 11.80% | 8.70% | 23.50% | 15.10% | 22.20% | 13.50% | 16.30% | 13.40% | 10.70% | 11.20% | 6.90% | 4.40% | 3.60% |
| | PGD | 17.00% | 100.00% | 11.10% | 8.70% | 20.10% | 12.90% | 20.20% | 11.20% | 14.90% | 13.00% | 11.90% | 11.50% | 5.90% | 3.50% | 3.40% |
| | DI-FGSM | 43.60% | 99.10% | 38.80% | 24.80% | 54.10% | 43.60% | 56.40% | 43.40% | 36.70% | 33.80% | 26.50% | 26.30% | 16.20% | 12.70% | 9.60% |
| | TI-FGSM | 21.50% | 91.90% | 24.80% | 18.90% | 32.70% | 30.10% | 35.20% | 25.60% | 20.60% | 18.60% | 13.60% | 15.50% | 14.60% | 16.00% | 11.90% |
| | MI-FGSM | 38.10% | 100.00% | 34.30% | 27.40% | 46.70% | 38.20% | 44.60% | 34.70% | 35.90% | 34.40% | 27.10% | 30.40% | 19.10% | 18.30% | 14.00% |
| | SINI-FGSM | 54.30% | 100.00% | 50.20% | 37.60% | 64.60% | 52.10% | 61.30% | 53.30% | 49.10% | 46.80% | 42.30% | 44.10% | 28.80% | 28.10% | 21.10% |
| | GNS-HFA (Ours) | **90.00%** | 99.60% | **87.50%** | **75.50%** | **92.10%** | **87.80%** | **90.30%** | **86.60%** | **85.10%** | **82.00%** | **78.60%** | **78.80%** | **68.60%** | **70.20%** | **61.60%** |
| CaiT-S/24 | TGR | 82.70% | 70.40% | 98.80% | 87.20% | 93.50% | 97.90% | 81.30% | 100.00% | 68.60% | 61.20% | 59.40% | 62.80% | 49.10% | 47.10% | 38.30% |
| | SSA | 77.30% | 73.50% | 88.40% | 83.30% | 87.70% | 88.80% | 77.30% | 97.50% | 75.60% | 73.60% | 72.60% | 73.00% | 69.10% | 68.20% | **66.10%** |
| | PNA | 59.70% | 53.80% | 82.70% | 65.40% | 76.20% | 82.30% | 59.50% | 94.10% | 49.20% | 45.40% | 41.70% | 44.50% | 31.80% | 28.20% | 22.90% |
| | BIM | 26.70% | 24.40% | 73.90% | 41.20% | 51.90% | 70.20% | 30.30% | 99.70% | 20.30% | 19.40% | 15.40% | 18.50% | 10.50% | 7.70% | 6.00% |
| | PGD | 25.70% | 23.60% | 67.80% | 36.90% | 45.00% | 64.70% | 27.20% | 99.60% | 20.70% | 16.80% | 15.70% | 18.20% | 7.30% | 5.90% | 4.70% |
| | DI-FGSM | 60.80% | 61.30% | 83.30% | 63.50% | 78.40% | 82.00% | 64.80% | 96.40% | 51.70% | 51.10% | 46.80% | 46.50% | 34.10% | 33.20% | 27.30% |
| | TI-FGSM | 36.20% | 40.00% | 61.10% | 42.40% | 59.00% | 61.50% | 47.90% | 87.80% | 30.40% | 31.30% | 24.00% | 26.10% | 26.80% | 26.90% | 22.60% |
| | MI-FGSM | 54.80% | 50.70% | 90.20% | 71.10% | 78.80% | 88.10% | 55.50% | 99.90% | 48.70% | 43.00% | 39.50% | 44.30% | 31.60% | 28.60% | 23.30% |
| | SINI-FGSM | 61.20% | 53.80% | 92.70% | 77.50% | 82.50% | 92.10% | 59.80% | 100.00% | 55.50% | 50.40% | 47.00% | 50.70% | 38.00% | 38.30% | 31.20% |
| | GNS-HFA (Ours) | **94.10%** | **87.70%** | **99.10%** | **95.90%** | **97.90%** | **98.90%** | **91.50%** | 100.00% | **84.70%** | **80.70%** | **81.70%** | **81.90%** | **74.20%** | **73.60%** | 64.60% |

Table 2: Performance comparison of ViT and CNN models with different $u$

| u | ViT | | | | | | | | CNN | | | | | | |
|---|---|---|---|---|---|---|---|---|---|---|---|---|---|---|---|
| | LeViT-256 | PiT-B | DeiT-B | ViT-B/16 | TNT-S | ConViT-B | Visformer-S | CaiT-S/24 | Inc-v3 | Inc-v4 | IncRes-v2 | ResNet-101 | Inc-v3-adv-3 | Inc-v3-adv-4 | IncRes-v2-adv |
| -1 | 73.50% | 69.90% | 91.50% | 99.70% | 85.80% | 91.90% | 71.50% | 91.70% | 65.50% | 61.70% | 56.80% | 61.80% | 53.30% | 53.10% | 45.30% |
| -0.8 | 74.40% | 69.40% | 91.90% | 99.80% | 86.50% | 92.20% | 71.20% | 91.10% | 66.00% | 62.20% | 57.30% | 61.90% | 52.80% | 53.30% | 46.20% |
| -0.6 | 74.40% | 70.10% | 92.50% | 99.80% | 86.60% | 91.70% | 71.60% | 91.70% | 66.40% | 62.10% | 57.90% | 61.90% | 54.60% | 53.70% | 45.80% |
| -0.4 | 74.80% | 70.80% | 92.50% | 99.90% | 86.70% | 92.60% | 71.70% | 92.00% | 66.00% | 62.80% | 57.90% | 61.30% | 54.00% | 54.20% | 47.20% |
| -0.2 | 74.80% | 70.00% | 92.70% | 99.90% | 87.10% | 92.10% | 72.30% | 92.20% | 66.80% | 64.30% | 59.20% | 62.20% | 53.50% | 54.70% | 47.80% |
| 0 | 76.30% | 70.30% | 92.40% | 99.90% | 87.60% | 92.30% | 72.00% | 92.10% | 67.90% | 63.30% | 59.10% | 62.90% | 52.90% | 54.30% | 47.20% |
| 0.2 | 76.00% | 70.50% | 92.70% | 99.90% | 87.70% | 93.20% | 72.90% | 92.20% | 67.80% | 64.10% | 58.20% | 63.30% | 53.50% | 54.50% | 47.50% |
| 0.4 | 75.40% | 70.70% | 93.20% | 99.90% | 88.20% | 92.90% | 73.00% | 92.00% | 67.80% | 64.20% | 58.20% | 63.50% | 53.90% | 54.70% | 47.70% |
| 0.6 | 75.50% | 70.40% | 93.60% | 99.90% | 88.00% | 93.10% | 72.80% | 92.30% | 67.80% | 64.10% | 58.90% | 63.50% | 54.70% | 55.00% | 47.30% |
| 0.8 | 75.70% | 69.60% | 93.20% | 99.90% | 86.70% | 93.10% | 73.50% | 92.50% | 67.20% | 63.10% | 59.30% | 63.70% | 54.40% | 53.90% | 47.10% |
| 1 | 75.70% | 70.30% | 93.10% | 99.90% | 88.40% | 92.60% | 74.10% | 92.30% | 68.30% | 63.90% | 58.90% | 63.00% | 54.50% | 54.90% | 47.80% |

**Parameter Setting:** We set the allowed deviation level $u$ to 1, the number of frequency samples explored $N$ to 20, the number of iterations $T$ to 10, and the perturbation magnitude $\epsilon$ to 16. Furthermore, we normalize the perturbation magnitude by $\epsilon/255$.

## 5.2 EXPERIMENTAL RESULT

As shown in Tab. 1, our method outperforms the existing methods both on ViT and CNN models. Specifically, our method demonstrated an overall improvement of 37.80% compared to all competitive baselines. In detail, it exhibited a 33.54% average improvement on ViT models and a 42.05% average improvement on CNN models. In comparison to the primary competing model TGR, our method exhibited an average improvement of 15.04%, with a 9.72% improvement on ViT models and a 20.35% improvement on CNN models. When compared to the best-performing transferable attack method SSA on CNNs, our method demonstrated an average improvement of 12.32%, with a 13.94% improvement on ViT models and a 10.71% improvement on CNN models.

## 5.3 ABLATION STUDY

**The effect of parameter $u$ on GNS-HFA** In the experiment, we fixed $N$ at 20 and $\epsilon$ at 16. The parameter $u$ was systematically varied within the range of -1 to 1, with increments of 0.2. It is evident in Tab. 2 that as $u$ approaches 1, GNS-HFA has an optimal performance. Nevertheless, it is worth noting that the overall impact on GNS-HFA's performance remains relatively modest.

Table 3: Performance comparison of ViT and CNN models with different $N$

| | ViT | | | | | | | | CNN | | | | | | |
|---|---|---|---|---|---|---|---|---|---|---|---|---|---|---|---|
| $N$ | LeViT-256 | PiT-B | DeiT-B | ViT-B/16 | TNT-S | ConViT-B | Visformer-S | CaiT-S/24 | Inc-v3 | Inc-v4 | IncRes-v2 | ResNet-101 | Inc-v3-adv-3 | Inc-v3-adv-4 | IncRes-v2-adv |
| 5 | 75.20% | 69.20% | 92.10% | 99.80% | 87.20% | 92.50% | 72.20% | 92.20% | 66.80% | 62.40% | 58.70% | 61.30% | 53.20% | 53.50% | 45.50% |
| 10 | 76.90% | 70.10% | 93.50% | 99.90% | 87.50% | 92.80% | 72.50% | 92.50% | 67.20% | 62.70% | 58.80% | 61.80% | 54.00% | 54.10% | 46.00% |
| 15 | 75.60% | 70.20% | 93.70% | 99.90% | 88.00% | 92.70% | 73.20% | 92.60% | 67.80% | 64.30% | 59.30% | 62.70% | 54.20% | 54.70% | 47.40% |
| 20 | 76.30% | 70.10% | 93.50% | 99.90% | 88.00% | 92.90% | 73.70% | 92.80% | 68.00% | 63.40% | 59.10% | 63.00% | 54.90% | 54.50% | 47.90% |

Table 4: Performance comparison of ViT and CNN models with different $\epsilon$

| | ViT | | | | | | | | CNN | | | | | | |
|---|---|---|---|---|---|---|---|---|---|---|---|---|---|---|---|
| $\epsilon$ | LeViT-256 | PiT-B | DeiT-B | ViT-B/16 | TNT-S | ConViT-B | Visformer-S | CaiT-S/24 | Inc-v3 | Inc-v4 | IncRes-v2 | ResNet-101 | Inc-v3-adv-3 | Inc-v3-adv-4 | IncRes-v2-adv |
| 8 | 40.90% | 39.20% | 66.30% | 97.90% | 59.40% | 65.90% | 40.40% | 67.90% | 35.80% | 32.40% | 27.70% | 32.80% | 25.20% | 24.60% | 20.50% |
| 16 | 76.30% | 70.10% | 93.50% | 99.90% | 73.70% | 92.90% | 88.00% | 92.80% | 68.00% | 63.40% | 59.10% | 63.00% | 54.90% | 54.50% | 47.90% |
| 24 | 88.00% | 83.20% | 97.80% | 100.00% | 97.40% | 97.80% | 87.80% | 95.40% | 82.40% | 81.20% | 76.60% | 80.00% | 70.70% | 70.90% | 63.50% |
| 32 | 94.50% | 90.80% | 98.70% | 100.00% | 97.80% | 99.40% | 98.60% | 93.40% | 88.90% | 86.60% | 85.10% | 86.10% | 81.30% | 79.00% | 74.70% |

We find that, regardless of the value of $u$, removing mild gradients always has a higher attack success rate than removing extreme gradients. This means that removing mild gradients can remediate the overfitting concern when the local surrogate ViT model trains adversarial examples, leading to the enhancement of the transferability of adversarial examples. Therefore, in this work, we consider mild gradients to exhibit a stronger correlation and have thus helped to improve the transferable adversarial attack success rate against the ViT models.

**The effect of parameter $N$ on GNS-HFA** In this experiment, we set $u$ to 1 and $\epsilon$ to 16, and varied $N$ at values of 5, 10, 15, and 20. As shown in Tab. 3, the performance of GNS-HFA consistently improves while $N$ increases. Notably, the optimal performance is achieved when $N$ is 20.

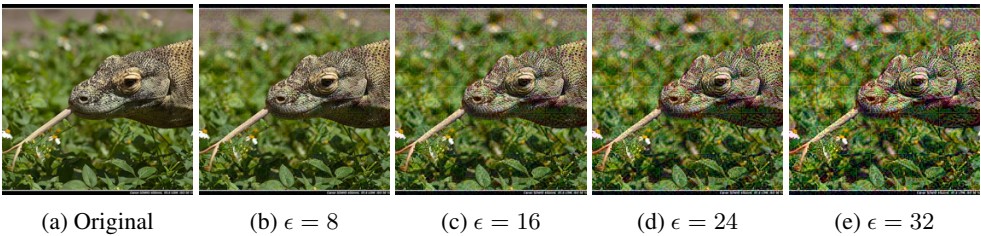

| (a) Original | (b) $\epsilon = 8$ | (c) $\epsilon = 16$ | (d) $\epsilon = 24$ | (e) $\epsilon = 32$ |

Figure 4: Adversarial attack sample at different $\epsilon$

**The effect of parameter $\epsilon$ on GNS-HFA** In the experiment, we set $u$ to 1 and $N$ to 20. We change the value of $\epsilon$ at 8, 16, 24, and 32. It can be observed that with the increase of $\epsilon$, there is a significant improvement in the performance of GNS-HFA, particularly evident in the ViT model, where the ASR surpasses 90% when $\epsilon$ is set to 32. However, this could be attributed to an excessive level of perturbation on the images. In Fig. 4, it is evident that as $\epsilon$ increases, the perturbation on the images becomes more pronounced. Nevertheless, at $\epsilon$ of 16, the perturbation remains relatively inconspicuous while still achieving a comparatively effective adversarial attack. Hence, the selection of $\epsilon$ can be customized for specific real-world requirements for varying degrees of attack effectiveness.

## 6 CONCLUSION

In this paper, we highlight the challenges of existing transferable attacks on ViTs, particularly in accurately locating and regularizing the gradients that lead to overfitting during the backpropagation process. Therefore, we present the Gradient Normalization Scaling and High-Frequency Adaptation method (GNS-HFA) method for fine-grained gradient editing and transferable attack direction adaptation, which significantly enhances the transferability of adversarial samples on ViTs. Extensive experiments conducted on various ViT variants and conventional CNN models demonstrate the superiority of our approach. We anticipate that this work could shed some light in this direction by providing the replication package publicly.

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

## A THE EFFECT OF REMOVING MILD GRADIENTS AND EXTREME GRADIENTS RESPECTIVELY ON THE CAIT-S/24 MODEL

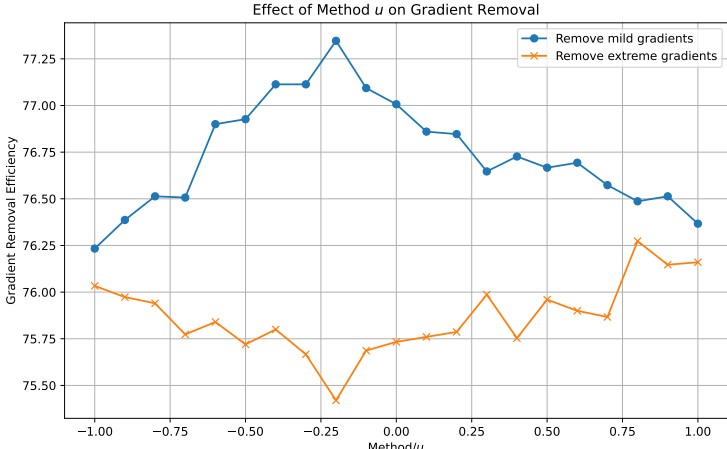

Figure 5: Removing mild gradients and extreme gradients respectively on the CaiT-S/24 model

As shown in Fig. 5, we constructed the adversarial samples on the CaiT-S/24 model by removing mild gradients and extreme gradients respectively, and calculated the average attack success rate after these adversarial samples were transferred to attack other ViT and CNN models. Different hyperparameters in the table adjust the division range of mild gradients. In transferable attacks, the transferability of adversarial samples can be largely affected by overfitting during local training, thus showing different attack success rates on the target models. Therefore, we believe that a higher average attack success rate represents a lower possibility of overfitting.

## B THE IMPACT OF GNS AND HFA ON SAMPLE TRANSFERABILITY

Table 5: Ablation study of GNS only, HFA only, both used and none used

| Method | ViT | | | | | | | | CNN | | | | | | | |
| | LeViT-256 | PiT-B | DeiT-B | ViT-B/16 | TNT-S | ConViT-B | Visformer-S | CaiT-S/24 | Inc-v3 | Inc-v4 | IncRes-v2 | ResNet-101 | Inc-v3-adv-3 | Inc-v3-adv-4 | IncRes-v2-adv | Average |
|---|---|---|---|---|---|---|---|---|---|---|---|---|---|---|---|---|
| NONE USED | 34.10% | 34.00% | 62.80% | 100.00% | 50.60% | 64.80% | 37.10% | 64.70% | 32.30% | 30.60% | 26.30% | 32.30% | 23.30% | 21.00% | 19.70% | 42.24% |
| GNS_ONLY | 64.40% | 58.60% | 88.90% | 99.90% | 78.30% | 89.40% | 62.60% | 87.50% | 54.80% | 51.00% | 42.90% | 49.00% | 38.10% | 38.80% | 33.20% | 62.49% |
| HFA_ONLY | 58.40% | 55.40% | 84.00% | 100.00% | 74.50% | 84.10% | 57.19% | 84.80% | 55.19% | 52.79% | 49.00% | 51.99% | 45.10% | 43.79% | 39.80% | 62.40% |
| BOTH | 76.30% | 70.10% | 93.50% | 99.90% | 88.00% | 92.90% | 73.70% | 92.80% | 68.00% | 63.40% | 59.10% | 63.00% | 54.90% | 54.50% | 47.90% | 73.20% |

We conducted ablation experiments for GNS only or HFA only in Tab. 5 with the same parameters in **Sec.5.1** of the main paper. The experimental results indicate that, compared to attack methods without either GNS or HFA (average success rate of 42.24%), both GNS and HFA play nearly equally crucial roles in enhancing the transferability of adversarial samples (averaging 62.49% and 62.40%, respectively). The strategy to combine GNS and HFA together yields the best algorithm performance (average 73.20%), demonstrating that gradient normalization and scaling for 'mild gradients', coupled with frequency-domain exploration, effectively improve the transferability of adversarial samples. The results of the ablation experiments align with our assumptions regarding the roles played by GNS and HFA.

