# OpenReview forum: "Enhancing Transferable Adversarial Attacks on Vision Transformers through Gradient Normalization Scaling and High-Frequency Adaptation"
_ICLR.cc/2024/Conference — ICLR 2024 poster_

### Official Review · Reviewer_Qpbi · 2023-10-29

**Soundness:** 2 fair
**Presentation:** 1 poor
**Contribution:** 2 fair
**Rating:** 6
**Confidence:** 4

**Summary:**

This paper proposes a method to increase the tranferrability of adversarial attacks across Transformer models. The key idea is to attenuate mild gradients and to do frequency adaptive perturbation of the input signal.

After rebuttal, I decided to increase my score to reflect the answers.

**Strengths:**

The comperhensive numerical results in the paper across a number of architectures and attack methods shows the benefits of the approach, consistently achieving very high transferrability scores. The method itself seems quite simple to implement.

**Weaknesses:**

This paper was written in a manner which made it very difficult for me to follow the exact approach. I have a number of questions below which I hope the authors will address. I am not up to date on the latest adversarial literature and therefore it may be that I have missed obvious ideas, but nevertheless I think the authors should write the paper for a general ICLR audience rather than adversarial sub-field experts. I am willing to re-visit my rating if the authors can provide satisfactory answers to the questions below, which are mostly to do with the very opaque setup in the paper.

**Questions:**

Questions:

1. What are "strong" and "mild" gradients? These terms are assumed to be understood by the reader, but never explicitly defined. At the least, one would expect some informal definition to give the reader some intuition.

2. The definition of "channels" in a Transformer model is unclear. For conv nets, it is obvious what this refers to. It seems to refer to the number of attention heads, but it was not too clear.

3. It is unclear why attentuating certain types of gradients ("mild") leads to better transferability. Is there an intuition that the authors can provide for this phenomenon?

4. The HFA method was quite unclear to me (and I guess it will be to many readers).

5. How are the GNS and HFA methods combined?

6. What exactly do the results in Table 1 show us? Is there one model on which the adversarial attacks were generated, and the remaining were the rates of success?

---

> ### Author Response · Authors · 2023-11-17
> **Official reply to Reviewer Qpbi (Part 1)**
>
> **Questions:**
>
> 1.Due to the internal structure differences between Vision Transformers (ViTs) and Convolutional Neural Networks (CNNs), traditional transferable attack methods designed for CNNs often yield suboptimal results when crafting adversarial samples to attack ViTs.
>
> The TGR method, as proposed by [1], attributes the limitation of adversarial sample transferability to the overfitting caused by ‘extreme gradients’ during the backpropagation of ViT models. Specifically, ‘extreme gradients’ are defined as tokens whose backpropagated gradient magnitudes rank in the top-$k$ or bottom-$k$ among all tokens, with $k$ being a hyperparameter.
>
> To reduce the variance in gradient backpropagation and enhance adversarial sample transferability, TGR regularizes the ‘extreme gradients’ corresponding to the top-$k$ or bottom-$k$ magnitudes.
>
> Our experiments, illustrated in **Figure 3 of the paper** and further validated in **Table 1 of the global comments**, demonstrate that, compared to the ‘extreme gradients’ addressed by TGR, small gradients, referred to as ‘mild gradients’, have more significant impacts leading to overfitting during gradient backpropagation.
>
> In this study, we employ the threshold $\mu+u * \sigma$ to differentiate between ‘mild gradients’ and ‘extreme gradients’, where $\mu$ represents the average value across $C$ channels, $\sigma$ represents the standard deviation across $C$ channels, and the hyperparameter $u$ defines the allowed deviation level. Gradients with magnitudes less than $\mu+u * \sigma$ are considered ‘mild gradients’. By adjusting the hyperparameter $u, we can alter the boundary for classifying ‘mild gradients’.
>
> On the one hand, our classification of ‘mild gradients’ implies lower gradient magnitudes compared to the ‘extreme gradients’ associated with top-$k$ or bottom-$k$ magnitudes. Thus, normalizing and scaling these ‘mild gradients’ causing overfitting does not significantly impact the success rate of attacks. On the other hand, ablation experiments in Section 5.3 of the manuscript demonstrate that our method is not highly dependent on the choice of the $u$ parameter. After normalizing and scaling the majority of ‘mild gradients’, our algorithm achieves outstanding performance, providing further evidence that ‘mild gradients’ are the primary contributors to overfitting. Since removing the vast majority of ‘mild gradients’ is sufficient, the algorithm's performance remains largely unchanged with variations in the parameter $u$.
>
> 2.We are pleased to address the reviewer's inquiries. In ViT, images are initially divided into fixed-size ‘patches’. Each patch is linearly mapped to a lower-dimensional vector, which serves as the input to the model. In this context, for ViT models employing convolution as Q (queries), K (keys), and V (values), the term ‘channel’ refers to the number of channels in the convolutional kernel. On the other hand, for ViT models utilizing fully connected layers, ‘channel’ denotes the number of heads in the multi-head attention mechanism.
>
> ​​3. Following we provide an intuitive explanation for the phenomenon of ‘mild gradients’ causing overfitting. Our goal is to train locally adversarial samples on the ViT model and transfer them to attack other ViT or CNN models effectively. It means that, we would like to see less overfitting occurring to the local surrogate ViT model during the training of adversarial samples. This may happen, if the model can learn largely with the gradients that are not model-specific and unstable. This ensures that transferable adversarial samples can effectively cross the decision boundaries of other ViT or CNN models, thereby misleading model decisions.
>
> Given the inherent structure differences between ViTs and CNNs, ViTs employ self-attention mechanisms to capture both global and local information in images. This mechanism makes the model highly sensitive to small perturbation in the input during the learning process. If gradients are too small, the update steps will be relatively small, causing the model to be sensitive to smaller perturbation in the training data. This may result in overfitting, subsequently affecting the quality of locally trained adversarial samples.
>
> On the other hand, ‘mild gradients’ may cause ViTs to get stuck in local optima during the learning process, making the model more prone to overfitting. For deeper ViT models, ‘mild gradients’ may also contribute to the vanishing gradient problem, preventing effective updates to the lower-level attention mechanisms thereby impacting the training process for adversarial samples.

---

> ### Author Response · Authors · 2023-11-17
> **Official reply to Reviewer Qpbi (Part 2)**
>
> 4.Due to the internal structure differences between ViT and CNN models, as illustrated in **Figure 2**, we have demonstrated that ViT, unlike CNN models, tends to focus more on high-frequency information. Therefore, it becomes essential to explore the high-frequency regions in the frequency domain to which ViT models are more sensitive, as these regions often contain crucial features influencing model decisions. Firstly, as shown in Equation 7, we utilize Discrete Cosine Transformation (DCT) [2] to convert the spatial information of the perturbed original image into frequency domain information. We introduce randomness through a normal distribution and use the mask constructed by Equation 6 to ensure the exploration of more high-frequency information. Secondly, we employ Inverse Discrete Cosine Transformation (IDCT) [2] to transform the explored high-frequency information back into spatial information. Subsequently, we perform frequency domain exploration for $N$ iterations. We calculate the average gradient of the $N$ samples after frequency domain exploration using Equation 8. The $sign$ function is then applied to determine the gradient update direction corresponding to the average gradient. Finally, we use High Frequency Adaptation (HFA) to guide and adjust the gradient ascent update process for adversarial samples. The gradient update direction obtained through HFA is tailored to the internal structure of ViT models, ensuring that each update moves the gradient in the most adversarial direction. It is worth noting that we have also demonstrated that GNS-HFA achieves optimal results on traditional CNN models.
>
> 5.As mentioned in the previous responses, we observed that in the backpropagation of ViT models, ‘mild gradients’ are more prone to overfitting compared to ‘extreme gradients’. Additionally, ‘mild gradients’ have lower gradient magnitudes compared to ‘extreme gradients’, so altering ‘mild gradients’ does not have a significant impact on the adversarial nature of the samples.
>
> We mitigate the occurrence of overfitting in backward propagation by normalizing and scaling ‘mild gradients’, thereby enhancing the transferability of adversarial samples trained on the local ViT model. This forms the basis of our Gradient Normalization Scaling (GNS) approach. Building upon GNS, and recognizing that ViT models emphasize high-frequency information more than CNN models, we aim to explore the high-frequency regions of ViT models.
>
> We further utilize High Frequency Adaptation (HFA) to guide and adjust the update process of adversarial samples by incorporating the explored high-frequency information. This ensures that the generated adversarial examples are more adaptable to the structure of the ViT model, and thus can more stably cross the decision boundary of the model in transferable attacks.
>
> Therefore, we first normalize and scale ‘mild gradients’ through GNS, and subsequently employ the high-quality gradient information obtained from GNS for frequency domain exploration. This process ultimately leads to the creation of highly transferable adversarial samples.
>
> We conducted additional ablation experiments for GNS or HFA in **Table 2** of the global comments. The experimental results indicate that, compared to attack methods without either GNS or HFA (average success rate of 42.24%), both GNS and HFA play nearly equally crucial roles in enhancing the transferability of adversarial samples (averaging 62.49% and 62.40%, respectively). Combining GNS and HFA yields the best algorithm performance (average 73.20%), demonstrating that gradient normalization and scaling for 'mild gradients', coupled with frequency-domain exploration, effectively improves the transferability of adversarial samples. The results of the ablation experiments align with our assumptions regarding the roles played by GNS and HFA.

---

> ### Author Response · Authors · 2023-11-17
> **Official reply to Reviewer Qpbi (Part 3)**
>
> 6.Thanks for highlighting the concern for result discussion.
>
> Table 1 serves as a commonly employed quantitative analysis of the success rates of transferable adversarial attacks, as also discussed in [3] and [4]. In the first column labeled ‘Surrogate Models’, various methods (such as TGR [1], SSA [4], etc., as indicated in the table) are used to train adversarial samples. These samples are then directly transferred to other ViT or CNN target models to assess the attack success rates. A higher attack success rate implies greater transferability. More information about the experimental settings can be found in [3], [4].
>
> We will revise the discussion of the experimental design in the new version accordingly.
>
> References:
>
> [1] Zhang, J., Huang, Y., Wu, W., & Lyu, M. R. (2023). Transferable Adversarial Attacks on Vision Transformers with Token Gradient Regularization. In Proceedings of the IEEE/CVF Conference on Computer Vision and Pattern Recognition (pp. 16415-16424).
>
> [2] Ahmed, N., Natarajan, T., & Rao, K. R. (1974). Discrete cosine transform. IEEE transactions on Computers, 100(1), 90-93.
>
> [3] Zhang, J., Wu, W., Huang, J. T., Huang, Y., Wang, W., Su, Y., & Lyu, M. R. (2022). Improving adversarial transferability via neuron attribution-based attacks. In Proceedings of the IEEE/CVF Conference on Computer Vision and Pattern Recognition (pp. 14993-15002).
>
> [4] Long, Y., Zhang, Q., Zeng, B., Gao, L., Liu, X., Zhang, J., & Song, J. (2022, October). Frequency domain model augmentation for adversarial attack. In European Conference on Computer Vision (pp. 549-566). Cham: Springer Nature Switzerland.

---

> > ### Comment · Reviewer_Qpbi · 2023-11-21
> > **My concerns are addressed.**
> >
> > Thank you for a detailed answer to my questions.

---

> > > ### Author Response · Authors · 2023-11-21
> > > **Thanks so much for your support**
> > >
> > > It is much appreciated for your constructive comments and the change of the score! It means a lot! Thank you.

---

### Official Review · Reviewer_6tAt · 2023-11-02

**Soundness:** 3 good
**Presentation:** 2 fair
**Contribution:** 2 fair
**Rating:** 5
**Confidence:** 3

**Summary:**

In order to enhance the transferability of adversarial attacks on ViTs, this paper introduces a novel gradient normalization scaling method for fine-grained gradient editing. After calculating the distribution of gradients, tanh is used for scaling the gradients that are considered as prone to cause overfitting and have minimal impact on attack capability. A high frequency adaptation method is proposed to explore the sensitivity of ViTs to adversarial attacks in different frequency regions, on the premise that ViTs shows different attention areas from CNNs in frequency. DCT transformation is conducted to obtain high frequency features, and then reverse transformation is put afterward to feed into the network for backpropagation. From the comparison of Attack Success Rates on ViT and CNN models, this work achieves better performance, enhancing the transferability of adversarial samples on ViTs.

**Strengths:**

1. This paper comes up with a novel problem space.
2. Utilizing normalization, or scaling is innovative.
3. Achieved better attack transferability performance than SOTA methods.

**Weaknesses:**

1.	Lack of soundness of mild gradients and extreme gradients.
2.	Normalization and scaling process is insufficiently demonstrated.
3.	Miss rationale of utilizing high frequency. Is frequency transformation for better ViTs or for better attacks?
4.	Ablation study is insufficient to evaluate the impact of your two components on the final method performance and verify their importance.
5.	Some sentences contain grammatical errors, such as missing subjects.

**Questions:**

1.	In your Abstract, after a one-sentence introduction to ViT, quickly talk about enhancing the transferability of adversarial attacks on ViTs is somewhat discontinuous. Adding an introduction to adversarial attacks in between would make it smoother. There are many more logical breaks like this.
2.	Structure illustration figure can be more detailed/comprehensible. Figure annotations could provide more explanation.
3.	There should be more explanations about how mild gradients and extreme gradients in Figure 3 react to your specific parameters (briefly introduce u here instead of later), and why mild gradients are the easiest to overfit. Besides, why is µ + u ∗ σ the watershed between mild and extreme gradients?
4.	Normalization and scaling seem to be one and the same.
5.	Give more arguments for tanh and frequency transformation.
6.	GNS-HFA is a combination, so does each part fit your hypothesis? What role do they play? Which one contributes more?

---

> ### Author Response · Authors · 2023-11-17
> **Official reply to Reviewer 6tAt (Part 1)**
>
> **Weaknesses:**
>
> 1.In **Section 4.1 of our paper**, we have demonstrated a distinction from TGR [1], which posits that ‘extreme gradients’ cause overfitting in the backpropagation process. Differently, we found that the small gradients, referred to as ‘mild’ gradients, are more prone to overfitting in backpropagation. Normalizing and scaling these ‘mild’ gradients have a relatively lower impact on sample attackability compared to TGR, which directly normalizes ‘extreme gradients’.
>
> 2.Experimental results on ViTs and CNNs models validate the effectiveness of our method to normalize and scale ‘mild gradients’ in enhancing the transferability of adversarial samples. In Section 4.1 of the paper, we provide a detailed explanation for the normalization and scaling process. As illustrated in **Figure 3 in our paper**, we observe that ‘mild gradients’ are more susceptible to overfitting in backpropagation compared to the ‘extreme gradients’ described by TGR.
>
> Therefore, we normalize 'mild gradients.' Subsequently, we employ the tanh activation function to map bias values. If there is a noticeable bias in the gradient information, it indicates the crucial role of relatively larger gradients in attack capability, which should not be eliminated. Hence, using the tanh activation function achieves adaptive scaling of gradients.
>
> 3.As depicted in **Figure 2 of the paper**, we visualized the frequency domain attribution map of the ViT model. Our observation reveals that, in contrast to traditional CNN models, the ViT model tends to focus more on high-frequency information. Therefore, we aim to guide and adjust the gradient update direction of adversarial samples through frequency domain transformation, to be able to construct adversarial samples with higher transferability for black-box attacks.
>
> 4.We conducted additional ablation experiments for GNS or HFA in **Table 2 of the global comments**. The experimental results indicate that, compared to attack methods without either GNS or HFA (average success rate of 42.24%), both GNS and HFA play nearly equally crucial roles in enhancing the transferability of adversarial samples (averaging 62.49% and 62.40%, respectively). The strategy to combine GNS and HFA together yields the best algorithm performance (average 73.20%), demonstrating that gradient normalization and scaling for 'mild gradients', coupled with frequency-domain exploration, effectively improves the transferability of adversarial samples. The results of the ablation experiments align with our assumptions regarding the roles played by GNS and HFA.
>
> 5.We will thoroughly proof-read the paper.
>
> **Questions:**
>
> 1.Currently, Vision Transformers (ViTs) have successfully adapted the self-attention mechanism from Transformers for the generation of high-quality images, playing an increasingly crucial role in computer vision tasks. However, akin to Convolutional Neural Networks (CNNs), ViTs are susceptible to the influence of adversarial samples, raising security concerns in real-world applications.
>
> As one of the most effective black-box attack methods, transferable adversarial attacks, given their ability to generate adversarial samples on surrogate models and directly transfer them to the target model for attacks without accessing the target model parameters, have become a notable threat. However, due to the distinct internal structures of ViTs and CNNs, adversarial samples constructed by traditional transferable attack methods can not be generalized to ViTS. Therefore, it is imperative to propose more effective transferability attack methods to unveil latent vulnerabilities in ViTs.
>
> We will refine the narrative structure of the abstract as discussed above.
>
> 2.We will revise the structure diagram for better clarity, and improve the representation of legends to provide readers with a clearer understanding of our methodology.

---

> ### Author Response · Authors · 2023-11-17
> **Official reply to Reviewer 6tAt (Part 2)**
>
> 3.We are willing to provide additional ablation experiments on other ViT models to demonstrate the phenomenon that 'mild gradients' are more prone to overfitting compared to 'extreme gradients'. The experiments are presented in **Table 1 of the global comments**.
>
> To address the question of why ‘mild gradients’ are more likely to cause overfitting in backpropagation, we conducted experiments on the original ViT model without GNS and HFA modules, evaluating the impact of ‘mild gradients’ and ‘extreme gradients’ on the stability of backward propagation.
>
> As shown in **Figure 2 in the paper**, removing ‘mild gradients’ results in a greater improvement in attack success rate compared to removing ‘extreme gradients’, indicating that ‘mild gradients’ are more prone to overfitting, thereby affecting the training of adversarial samples on the surrogate model.
>
> An intuitive explanation for the increased susceptibility of ‘mild gradients’ to overfitting is that, in training highly transferable adversarial samples on ViT models for subsequent attacks on other ViT or CNN models, we aim for minimal overfitting to occur on the local surrogate ViT model. In other words, the model should learn gradients that are not model-specific and unstable to ensure the transferability of samples across decision boundaries of other ViT or CNN models, thereby misleading model outputs.
>
> Due to the differing internal structures of ViTs and CNNs, ViTs employ self-attention mechanisms to capture both global and local information in images. This mechanism renders the model highly sensitive to subtle perturbation in inputs during the training process. If gradients are too small, the update step is relatively small, making the model excessively sensitive to small changes in the training data, thus making overfitting more likely and affecting the quality of locally trained adversarial samples.
>
> On the other hand, ‘mild gradients’ may lead ViTs to fall into local minima during the learning process, making the model more prone to overfitting. For deeper ViT models, ‘mild gradients’ may also contribute to the vanishing gradient problem, preventing effective updates to the lower-level attention mechanisms and thereby affecting the training process of adversarial samples.
>
> Regarding the thresholding between ‘mild gradients’ and ‘extreme gradients’, we propose using $\mu+u * \sigma$ for this purpose, where $\mu$ represents the average value across $C$ channels, $\sigma$ represents the standard deviation across $C$ channels, and the hyperparameter $u$ denotes the allowed deviation level.  We consider gradient values smaller than $\mu+u * \sigma$ as ‘mild gradients’, which should be normalized and scaled. By adjusting the hyperparameter $u$, we can correspondingly adjust the range of ‘mild gradients’. It is worth noting that the ablation experiments on the parameter $u$ in Section 5.3 demonstrate that our method is not heavily dependent on the choice of $u$. After normalizing and scaling the majority of ‘mild gradients’, our algorithm achieves excellent performance, providing additional evidence that ‘mild gradients’ are the primary cause of overfitting. Since removing the vast majority of ‘mild gradients’ is sufficient, the algorithm's performance remains largely unchanged with variations in the parameter $u$.
>
> 4.We would like to clarify that normalization tends to scale gradient values to be within the range of 0 and 1, while scaling does not have such a restriction. The purpose of scaling is to reduce the impact of this portion of features on backpropagation by making the corresponding values smaller.
>
> 5.We are pleased to provide the relevant parameters for $tanh$ activation function and frequency domain transformation. Actually, they can be found in the replication package. We use the parameter $self.rho$ to control the range of exploration during frequency domain transformation, and we set it to 0.5. Larger values of $self.rho$ indicate a larger exploration range during frequency domain transformation.
>
> 6.Please find our detailed response to Weakness #4.
>
> [1] Zhang, J., Huang, Y., Wu, W., & Lyu, M. R. (2023). Transferable Adversarial Attacks on Vision Transformers with Token Gradient Regularization. In Proceedings of the IEEE/CVF Conference on Computer Vision and Pattern Recognition (pp. 16415-16424).

---

> > ### Author Response · Authors · 2023-11-23
> > **Request for reconsideration of the score**
> >
> > Dear Reviewer 6tAt,
> >
> > We extend our sincere gratitude for your dedication and comment to our paper. We fully comprehend the demanding workload and time constraints you have contributed.
> >
> > Nevertheless, we would greatly appreciate the availability of **continuous discussion and the possible reconsideration of your score** in light of our **rebuttal** and **the extended experimental results as requested**. This feedback is particularly valuable as it reflects a year-long commitment from our group.
> >
> > We genuinely thank you for your review and reconsideration.
> >
> > Best Regards,
> >
> > The Submission5069 Authors

---

### Official Review · Reviewer_98px · 2023-11-03

**Soundness:** 3 good
**Presentation:** 3 good
**Contribution:** 2 fair
**Rating:** 5
**Confidence:** 4

**Summary:**

In this submission, the authors proposed a gradient normalization scaling and high frequency adaptation for vision transformers. Specifically, the authors proposed to improve the generalization ability of ViTs by using gradient normalization. Moreover, the authors proposed a high frequency adaptation approach to guide the back-propagation in ViTs. Experimental results on several public datasets for adversarial attack have illustrated the effectiveness of the proposed method.

**Strengths:**

1. The paper is easy to follow.
2. The idea is well motivated and presented.

**Weaknesses:**

The contribution is marginal, since the gradient normalization was demonstrated in [1], please discuss the major differences.

[1] Wu, Y. L., Shuai, H. H., Tam, Z. R., & Chiu, H. Y. (2021). Gradient normalization for generative adversarial networks. In Proceedings of the IEEE/CVF International Conference on Computer Vision (pp. 6373-6382).

**Questions:**

Please conduct ablation studies using only GNS or HFA to demonstrate the effectiveness of those two methods.

---

> ### Author Response · Authors · 2023-11-17
> **Official reply to Reviewer 98px (Part 1)**
>
> **Weaknesses:**
>
> Thanks for the thorough review and insightful comment on our work.
>
> We would like to clarify that, in comparison to ‘Gradient normalization for generative adversarial networks’ [1], our approach is different, as well as the aim.
>
> Firstly, due to the difference in internal model structure between Vision Transformers (ViTs) and Convolutional Neural Networks (CNNs), conventional transferable attack methods that are effective on CNNs can not be generalized to ViTs. In particular, TGR [2] has presented that such limitations hindering the transferability of adversarial samples arise from the overfitting issue during the backpropagation process of ‘extreme gradients’ in ViT models.
>
> Specifically, a token's back-propagated gradient is considered extreme if it ranks in the top-$k$ or bottom-$k$ gradient magnitudes among all tokens, where $k$ is a hyperparameter.
>
> Therefore, to mitigate the variance in gradient backpropagation and enhance the transferability of adversarial samples, TGR regularizes the ‘extreme gradients’ associated with the top-k or bottom-k tokens [2]. Through empirical result analysis (Figure 3 in our paper), we demonstrate that, compared to the ‘extreme gradients'’ addressed by TGR, small gradients, referred to as ‘mild gradients’, also have a more significant impact on mitigating the overfitting issue during gradient backpropagation.
>
> An intuitive interpretation of this phenomenon is that, for Vision Transformers (ViTs), self-attention mechanisms are utilized to capture both global and local information within images. This mechanism renders the model highly sensitive to subtle variations in the input during the training process. If gradients become extremely small, the updating steps are relatively diminished, causing the model to be excessively sensitive to small perturbation in the training data. Consequently, this heightened sensitivity can lead to overfitting, impacting the quality of locally-trained adversarial samples.
>
> On the other hand, ‘mild gradients’ may lead ViTs to converge to local optimal during the training process, making the model more susceptible to overfitting. Additionally, for a deeper ViT model, ‘mild gradients’ might exacerbate the issue of gradient vanishing, rendering the lower-level attention mechanisms ineffective in receiving meaningful updates and consequently affecting the training process of adversarial samples. Further quantitative experiments validating these assertions are provided in **Table 1 of the global comment**.
>
> We aim to mitigate the overfitting issue during backpropagation by normalizing and scaling ‘mild gradients’, thereby enhancing the transferability of adversarial samples locally trained on Vision Transformer (ViT) models. This forms the basis of our work, Gradient Normalization Scaling (GNS).
>
> Building upon GNS, and recognizing ViT models' emphasis on high-frequency information over Convolutional Neural Network (CNN) models, we introduce High Frequency Adaptation (HFA). HFA explores the high-frequency regions of ViT models, utilizing the gleaned high-frequency information to adjust the update process of adversarial samples. This ensures that the generated adversarial examples are more adaptable to the structure of the ViT model, and thus can more stably cross the decision boundary of the model in transferable attacks. Consequently, we utilize the refined gradient information obtained after applying GNS for frequency-domain exploration, ultimately generating adversarial samples with heightened transferability.
>
> We will include the discussion with [1] in our revision, to provide a more comprehensive analysis.

---

> ### Author Response · Authors · 2023-11-17
> **Official reply to Reviewer 98px (Part 2)**
>
> **Questions:**
>
> We have conducted additional ablation experiments for GNS or HFA in **Table 2 in the global comments**. The experimental results indicate that, compared to attack methods without either GNS or HFA (average success rate of 42.24%), both GNS and HFA play nearly equally crucial roles in enhancing the transferability of adversarial samples (averaging 62.49% and 62.40%, respectively). The strategy of combining GNS and HFA yields the best algorithm performance (average 73.20%), demonstrating that gradient normalization and scaling for ‘mild gradients’, coupled with frequency-domain exploration, effectively improves the transferability of adversarial samples. The results of the ablation experiments align with our assumptions regarding the roles played by GNS and HFA.
>
> Reference:
>
> [1] Wu, Y. L., Shuai, H. H., Tam, Z. R., & Chiu, H. Y. (2021). Gradient normalization for generative adversarial networks. In Proceedings of the IEEE/CVF International Conference on Computer Vision (pp. 6373-6382).
>
> [2] Zhang, J., Huang, Y., Wu, W., & Lyu, M. R. (2023). Transferable Adversarial Attacks on Vision Transformers with Token Gradient Regularization. In Proceedings of the IEEE/CVF Conference on Computer Vision and Pattern Recognition (pp. 16415-16424).

---

> ### Comment · Reviewer_6tAt · 2023-11-21
>
> I have read the responses. Generally, the authors tried their best to explain the confusing points in the submission, and part of points can be supported. Yet, it seems that the paper should undergo a major revision for consideration of publication. Hence, I remain the original rating.

---

> > ### Author Response · Authors · 2023-11-21
> > **Response to the rating**
> >
> > Dear Reviewer 6tAt,
> >
> > We appreciate your time and consideration of our paper.
> >
> > As per requested, we have added the **additional experiments** in **Table 1** and **Table 2** to better clarify and support our discussion.
> >
> > Could you inform us if there is anything specific that requires revision? Thank you!
> >
> > Best,
> > The authors

---

### Author Response · Authors · 2023-11-17
**Experiments in global comments**

Table 1. Removing *mild gradients* and *extreme gradients* respectively on the CaiT-S/24 model

| $u$  | Remove mild gradients | Remove extreme gradients |
|------|:-----------------------:|--------------------------:|
| -1   | 76.23333              | 76.03333                 |
| -0.9 | 76.38667              | 75.97333                 |
| -0.8 | 76.51333              | 75.94                   |
| -0.7 | 76.50667              | 75.77333                 |
| -0.6 | 76.9                 | 75.84                   |
| -0.5 | 76.92667             | 75.72                   |
| -0.4 | 77.11333             | 75.8                    |
| -0.3 | 77.11333             | 75.66667                |
| -0.2 | 77.34667             | 75.42                   |
| -0.1 | 77.09333             | 75.68667                |
| 0    | 77.00667             | 75.73333                |
| 0.1  | 76.86                | 75.76                   |
| 0.2  | 76.84667             | 75.78667                |
| 0.3  | 76.64667             | 75.98667                |
| 0.4  | 76.72667             | 75.75333                |
| 0.5  | 76.66667             | 75.96                   |
| 0.6  | 76.69333             | 75.9                    |
| 0.7  | 76.57333             | 75.86667                |
| 0.8  | 76.48667             | 76.27333                |
| 0.9  | 76.51333             | 76.14667                |
| 1    | 76.36667             | 76.16                   |

As shown in Table 1, we constructed the adversarial samples on the CaiT-S/24 model by removing mild gradients and extreme gradients respectively, and calculated the average attack success rate after these adversarial samples were transferred to attack other ViT and CNN models. Different hyperparameters $u$ in the table adjust the division range of mild gradients. In transferable attacks, the transferability of adversarial samples can be largely affected by overfitting during local training, thus showing different attack success rates on the target models. Therefore, we believe that a higher average attack success rate represents a lower possibility of overfitting.

We find that, *no matter what the value of $u$ is, removing mild gradients always has a higher attack success rate than removing extreme gradients*. This means that removing mild gradients can remediate the overfitting concern when the local surrogate ViT model trains adversarial examples in comparison to removing extreme gradients, leading to the enhancement of the transferability of adversarial examples. Therefore, in this work, we consider mild gradients exhibit stronger correlation and have thus helped to improve the transferable adversarial attack success rate against the ViT models.

Table 2. Ablation experiments of GNS only, HFA only, both used and none used

| Method         | LeViT-256 | PiT-B  | DeiT-B  | ViT-B/16 | TNT-S | ConViT-B | Visformer-S | CaiT-S/24 | Inc-v3 | Inc-v4 | IncRes-v2 | ResNet-101 | Inc-v3-adv-3 | Inc-v3-adv-4 | IncRes-v2-adv | Average |
|----------------|:-----------:|:--------:|:---------:|:----------:|:-------:|:----------:|:-------------:|:-----------:|:--------:|:--------:|:-----------:|:------------:|:-------------:|:--------------:|:---------------:|---------:|
| None used      | 34.10%    | 34.00% | 62.80%  | 100.00%  | 50.60% | 64.80%   | 37.10%      | 64.70%    | 32.30% | 30.60% | 26.30%    | 32.30%     | 23.30%      | 21.00%       | 19.70%        | 42.24%  |
| GNS_ONLY       | 64.40%    | 58.60% | 88.90%  | 99.90%   | 78.30% | 89.40%   | 62.60%      | 87.50%    | 54.80% | 51.00% | 42.90%    | 49.00%     | 38.10%      | 38.80%       | 33.20%        | 62.49%  |
| HFA_ONLY       | 58.40%    | 55.40% | 84.00%  | 100.00%  | 74.50% | 84.10%   | 57.19%      | 84.80%    | 55.19% | 52.79% | 49.00%    | 51.99%     | 45.10%      | 43.79%       | 39.80%        | 62.40%  |
| BOTH           | 76.30%    | 70.10% | 93.50%  | 99.90%   | 88.00% | 92.90%   | 73.70%      | 92.80%    | 68.00% | 63.40% | 59.10%    | 63.00%     | 54.90%      | 54.50%       | 47.90%        | 73.20%  |

Here LeViT-256, PiT-B, DeiT-B, ViT-B/16, TNT-S, ConViT-B, Visformer-S and CaiT-S/24 are **ViT models**; Inc-v3, Inc-v4, IncRes- v2, ResNet-101, Inc-v3-adv-3, Inc-v3-adv-4 and IncRes-v2-adv are **CNN models**.

---

### Meta-Review · Area_Chair_SpfK · 2023-12-06

**Metareview:**

The paper focuses on improving transferable adversarial attacks on ViTs. In the first section, they identify that "mild gradients" and not "extreme gradients" are responsible for the lack of reduced attack transferability. Using this intuition, they propose a Gradient Normalization Scaling method, wherein the mild gradients are scaled. In the second section, they note that ViT seems to utilize high frequency features (though this isn't proven or cited) and use this to compute an attack which focuses on high frequency components. Based on a fairly extensive evaluation, they show greater success than baseline techniques on a number of surrogate and target models and also provide an ablation of several of the parameters.

**Justification For Why Not Higher Score:**

The paper has issues with both presentation (in terms of the writing) and also clear proof for each of their claims (e.g. ViT's focus on high frequency, visualizations of the effect of each of their components on the attacks, etc.)

**Justification For Why Not Lower Score:**

I think this paper can offer some value to the community. The technique seems to clearly outperform baselines and there are two clear intuitions.

---

### Decision · Program_Chairs · 2024-01-16

Accept (poster)